# Location of Pathogen Inoculum in the Potting Substrate Influences Damage by *Globisporangium ultimum*, *Fusarium culmorum* and *Rhizoctonia solani* and Effectiveness of Control Agents in Maize Seedlings

Eckhard Koch [1], Petra Zink [1], Tanja Bernhardt [1], Tim Birr [2] and Ada Linkies [1,*]

[1] Julius Kühn-Institute, Federal Research Centre for Cultivated Plants, Institute for Biological Control, 69221 Dossenheim, Germany; eckhard.koch@julius-kuehn.de (E.K.); pazink@freenet.de (P.Z.); tanja.bernhardt@julius-kuehn.de (T.B.)

[2] Department of Plant Diseases and Crop Protection, Institute of Phytopathology, Faculty of Agricultural and Nutritional Science, Christian-Albrechts-University of Kiel, 24118 Kiel, Germany; t.birr@phytomed.uni-kiel.de

* Correspondence: ada.linkies@julius-kuehn.de; Tel.: +49-39-4647-4960

**Abstract:** The aim of this study was to determine the impact of the location of the pathogen inoculum on damage caused by *Globisporangium* (syn. *Pythium*) *ultimum*, *Fusarium culmorum* and *Rhizoctonia solani* in pot tests with maize. For this purpose, pathogen inoculum was added to potting substrate, and the resulting mix was used to fill the whole pot volume, the upper half, or the lower half of pots. The remaining volume was filled with non-inoculated substrate. In a second experimental approach, maize seeds were germinated in non-inoculated potting substrate and the seedlings were transferred to inoculated substrate. The seeds were untreated, treated with the chemical thiram, or treated with a bacterial or a fungal biocontrol agent. With each of the pathogens, the damage to the developing maize seedlings was the strongest when the seeds germinated in the inoculated potting substrate. When only the roots were in contact with the inoculum, there was limited damage by *R. solani* and *F. culmorum,* and no damage by *G. ultimum*. This implies that in experiments with artificial inoculation, the seeds should always be in immediate contact with the inoculum if a strong pathogenic effect is desired. Conversely, seed treatments must, in the first place, be able to protect the spermosphere, while the requirement to protect the roots at a distance from the seed seems to depend on the pathogen.

**Keywords:** maize; seedling diseases; artificial inoculation; spermosphere; soilborne infection

## 1. Introduction

Seeds and seedlings of many field and vegetable crops are prone to attack by fungi causing diseases such as seed decay, damping off and seedling blight. The inciting pathogens are either located in or on the seed and are activated as the seed starts to germinate (seedborne pathogens), or they reside in the soil, where they often survive in the form of specialized resting structures that germinate in response to stimuli released from germinating seeds or plant roots (soilborne pathogens). Common to both is that the degree of infection and disease severity is generally directly dependent on the soil temperature and soil water content prevailing during germination and seedling development. Other factors determining the speed of plant development, such as soil texture, sowing time and sowing depth, also have an impact [1].

Among other measures, such as the use of pathogen-free seeds and the adoption of appropriate cropping practices, seed treatment with chemical fungicides [2] to enhance germination and vigor is probably the most practiced strategy to reduce losses from seed- and soilborne fungal pathogens. In Europe, some alternative seed treatments are currently on the market, namely, microbial agents [3] and physical methods, such as the use of

hot water, aerated steam [4] and treatment with low-energy electrons [5]. Compared to chemical seed treatment, the use of these alternatives is still limited to certain countries, regions and crops. There is, however, great interest and demand to explore the potential of these sustainable alternatives [6] and to develop new agents and combinations [7].

In maize, *Globisporangium ultimum* (syn. *Pythium ultimum*), several species of *Fusarium* [8,9] and different *Rhizoctonia solani* anastomosis groups (AGs) and subgroups are considered to be the most important pathogens causing seedling diseases. *Rhizoctonia solani* causes rot of lateral and crown roots of maize [10]. *Rhizoctonia solani* AG 2-2IIIB is the inciting pathogen of late crown and root rot in sugarbeet, but also causes root rot of maize [11]. As a causal agent of maize seedling diseases, *R. solani* has received relatively little attention compared to *G. ultimum* and fusaria. In greenhouse experiments, the infection of maize seedlings by *R. solani* led to dramatic reductions in root length, volume and surface area [12].

Pot tests performed in greenhouses or growth rooms lend themselves particularly well to work with seedling diseases and are frequently employed in screenings for new seed treatments and in experiments aimed at characterizing the performance of seed treatment agents and methods. Compared to field testing, greenhouse tests have the advantage that the growing conditions—including the temperature and water potential of the potting substrate—are controlled, the time between sowing and the appearance of disease symptoms is generally short, and a larger number of candidate agents can be tested simultaneously. Further, the infection conditions can be manipulated by the choice of the seed material (healthy, naturally infected or artificially inoculated) and the growth substrate (untreated, pasteurized or artificially inoculated). Artificial inoculation of the potting substrate is generally used to mimic disease pressure from soilborne pathogens. The outcome of such experiments is primarily dependent on the kind, concentration and location of the inoculum. Inocula used in pot tests with maize were fungal propagules [13,14] or intact or milled cereals colonized by the pathogens [12,15]. In these studies, the inoculum was mixed homogeneously into the potting substrate, while in others, it was applied as a layer [16,17]. However, systematic comparisons between different inoculation methods are apparently seldom made, or the results of such experiments may not be published once a suitable method of inoculation has been identified.

Here, we report on pot tests with maize that compared the pathogenic effect of inoculation with *G. ultimum*, *F. culmorum* and *R. solani* in relation to the location of the inoculum. The results show that the location of the inoculum has an impact on the resulting infection pressure and that the response depends on the pathogen employed.

## 2. Materials and Methods

### 2.1. Maize Seeds

The experiments were performed with the commercial hybrid maize seed variety Likeit (Deutsche Saatveredelung AG, Lippstadt, Germany).

### 2.2. Fungal Pathogens and Preparation of Inocula

The pathogenic fungi used were *Fusarium culmorum* VIII18, *Rhizoctonia solani* AG2-2 IIIB and *Globisporangium ultimum* (syn. *Pythium ultimum*) CC-3. *Fusarium culmorum* and *R. solani* were maintained on potato dextrose agar (PDA; Sigma-Aldrich, Schnelldorf, Germany), and *G. ultimum* was cultivated on Czapek agar (Czapek-Dox Broth (Sigma-Aldrich) solidified with 16 g agar/L). For inoculation with *F. culmorum*, millet seeds colonized with the pathogen were used. To 100 g autoclaved millet seeds in 1 L Erlenmeyer flasks, 30 mL conidial suspension ($1.5 \times 10^4$ conidia/mL) was added. The flasks were incubated at 20 °C in darkness for three days and agitated once a day to prevent the formation of clots. *Globisporangium ultimum* and *R. solani* were grown on seeds of buckwheat [18] and pearl barley, respectively. The substrates were filled into 1 L Erlenmeyer flasks with 100 g per flask. After autoclaving, ten agar discs (10 mm diameter) taken from actively growing cultures of the pathogens on PDA and 50 mL of sterile distilled water were added to each

flask. The flasks with *G. ultimum* were incubated at 20 °C in darkness for 5 days, with mixing after 2 days of incubation to ensure even growth. *R. solani* was cultivated for 6 days and flasks were shaken once at the end of the incubation period.

### 2.3. Seed Treatment

*Pseudomonas chlororaphis aurantiaca* BI 7439 was grown in shake cultures in 300 mL Erlenmeyer flasks filled with 50 mL tryptic soy broth (TSB; Merck, Darmstadt, Germany). The medium was inoculated with bacteria from TSA plates (TSB solidified with 16 g agar/L), the flasks were placed on a rotary shaker for 48 h (150 rpm, 50 mm deflection) at 25 °C and the resulting suspensions were used for seed inoculation.

*Trichoderma* sp. BI 7376 was cultured for 7 days on PDA plates at 20 °C in darkness. To sporulating cultures, 10 mL sterile distilled water with 0.0125% Tween 80 was added, and release of conidia was supported by agitating with a spatula. The concentration of conidia was adjusted to $1 \times 10^8$/mL with a hemocytometer, and 1% methylcellulose was dissolved in the suspension.

Seed inoculation was performed by placing maize kernels in the bacterial or conidial suspensions for 15 min, respectively. The kernels were then separated from the liquid by passing through a sieve, dried overnight in a laminar flow hood and sown the following day. The chemical seed treatment Thiram SC 700 (686 g/L thiram) was applied at a rate corresponding to 4 g active ingredient per kg of seed, as described in [7].

### 2.4. Growth Chamber Trials

The experiments were performed in $8 \times 8$ cm plastic pots in a pre-heated (48 h at 60 °C) potting substrate as described previously [7], with the following modifications. The pathogen inocula described above were evenly mixed into the potting substrate at concentrations of 0.5% (*F. culmorum*), 1% (*R. solani*) or 1.5% (*G. ultimum*), respectively. For each pathogen, the effect of the inoculum was evaluated at three different positions. This was achieved by (1) filling the entire pot volume with inoculated substrate, or (2) filling the lower half of pots with non-inoculated substrate and placing inoculated substrate above, or (3) filling the lower half of pots with inoculated substrate and adding non-inoculated substrate on top. The seeds sown in each of the three variants were either untreated (=pathogen control), treated with thiram or treated with *P. chlororaphis* BI 7439 (in experiments with *G. ultimum* and *F. culmorum*) or *Trichoderma* sp. BI 7376 (in experiments with *R. solani*). Pots filled entirely with non-inoculated substrate and sown with non-inoculated seeds served as healthy controls.

In a second experimental approach, the pots were half filled with non-inoculated substrate, and a piece of mesh fabric (approx. $20 \times 20$ cm, mesh size 2 mm) was placed over the surface. The pots were then filled up with non-inoculated substrate, resulting in two layers of non-inoculated substrate separated by the mesh fabric, and maize seeds were sown in the upper layer. After 5 days, the upper layer was transferred to pots half filled with non-inoculated substrate (control) or with substrate inoculated with *G. ultimum*, *F. culmorum* or *R. solani*, respectively.

Each treatment consisted of 5 pots with 5 seeds each. The experiments were repeated at least once. After sowing, the pots were covered with a layer (approx. 1 cm thick) of vermiculite, and their weight was recorded. They were then placed in a randomized design in a plant growth room at 20 °C and 50–70% relative humidity and covered loosely with an opaque plastic sheet. The latter was removed after 5 days and the pots were exposed to 16 h of light from fluorescent lamps. The pots were watered every second day by re-adjusting their weight to the original weight plus 10 g. Two weeks after sowing, the number of plants per pot was recorded, the plants were cut at the crown, and the plant dry weight in each pot was determined after incubation for 48 h at 60 °C.

### 2.5. Statistical Analysis

Data were statistically analyzed with the software SAS Studio 3.8. The results of the experiments were compared by using a generalized linear model—glim-mix procedure (GLMM; $p < 0.05$).

## 3. Results

### 3.1. Influence of Inoculum Location on Damage by Globisporangium ultimum

In the experiments with *G. ultimum*, the number of maize seedlings present two weeks after sowing tended to be reduced compared to the healthy control, both in the pots with even distribution of the inoculum in the entire potting substrate (variant 1) and in the pots with inoculum in the upper half of the pots (variant 2) (Figure 1a). None of these reductions, however, were statistically significant. Seed treatment with thiram and *Pseudomonas chlororaphis* also had no significant effect. No reduction in plant number was observed when the inoculum was only present in the lower half of the pots (variant 3). A similar pattern was observed regarding the plant dry weight per pot (Figure 1b). The latter was also unaffected in the pots of variant 3, but compared to the healthy control, it was significantly reduced in the pots of variant 1 (inoculum evenly distributed) and 2 (inoculum only in upper half of pots).

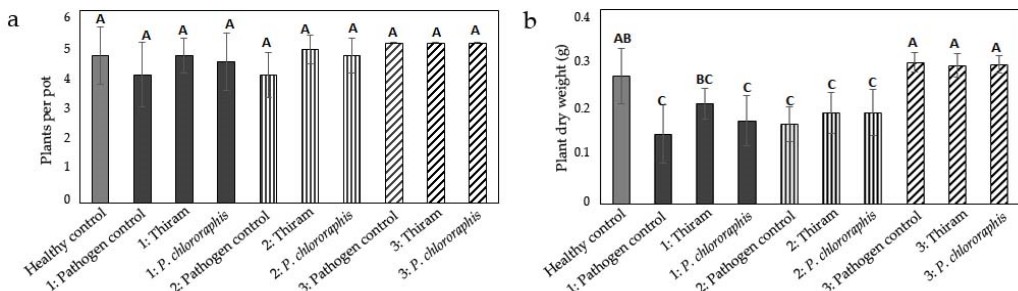

**Figure 1.** Effect of position of inoculum of *Globisporangium ultimum* on (**a**) number and (**b**) biomass of maize seedlings and on efficacy of seed treatment with thiram or a suspension of *Pseudomonas chlororaphis aurantiaca* BI 7439. Variant 1: inoculum of *G. ultimum* distributed homogenously in the entire pot volume; variant 2: inoculum of *G. ultimum* present only in the upper half of pots; variant 3: inoculum of *G. ultimum* present only in the lower half of pots. Means and standard deviation of five pots per treatment with five maize seeds each. Different letters above bars indicate statistically significant differences between the treatments (GLMM; $p < 0.05$).

### 3.2. Influence of Inoculum Location on Damage by Fusarium culmorum

Inoculation of the potting substrate with *F. culmorum* caused a significant reduction in the plant number when the inoculum was evenly distributed in the pots (variant 1) and when it was present in the upper half of the pots (variant 2) (Figure 2a). Seed treatment with thiram and *P. chlororaphis* appeared to be very effective, as the plant numbers were not different from the healthy controls. In pots with inoculum present only in the lower half (variant 3), a reduction in the number of plants was not observed.

The plant dry weight was much more affected than the plant number (Figure 2b). Compared to uninoculated substrate, it was drastically reduced in the pots with even distribution of the inoculum (variant 1) and in pots with inoculum present in the upper half (variant 2). In these variants, seed treatment with thiram and *P. chlororaphis* caused a significant increase in plant dry weight, although the effect was much lower than on plant number, and the resulting dry weight was still clearly lower than in the healthy control. The plant dry weight of the pathogen control of variant 3 (inoculum only in lower half of pots) was significantly reduced compared to the healthy control by about 23%, which was, however, much less than in variants 1 and 2, where the reduction was more than 95%.

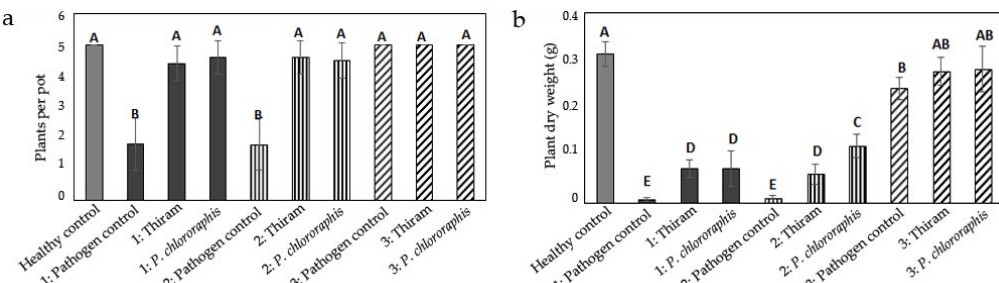

**Figure 2.** Effect of position of inoculum of *Fusarium culmorum* on (**a**) number and (**b**) biomass of maize seedlings and on efficacy of seed treatment with thiram or a suspension of *Pseudomonas chlororaphis aurantiaca* BI 7439. Variant 1: inoculum of *F. culmorum* mixed homogenously in the entire pot volume; variant 2: inoculum of *F. culmorum* present only in the upper half of pots; variant 3: inoculum of *F. culmorum* present only in the lower half of pots. Means and standard deviation of five pots per treatment with five maize seeds each. Different letters above bars indicate statistically significant differences between the treatments (GLMM; *p* < 0.05).

### 3.3. Influence of Inoculum Location on Damage by Rhizoctonia solani

Compared to the healthy control, inoculation of the potting substrate with *R. solani* caused a reduction in the number of plants in variant 1 only (Figure 3a), whereas the plant dry weight was significantly reduced in all variants and treatments (Figure 3b). With about −70%, the reduction was most pronounced in the pathogen controls of variant 1 and variant 2, and significantly larger than in the pathogen control of variant 3 (−46%) (Figures 3b and S1). Seed treatment with thiram caused a significant increase in plant dry weight over the pathogen control in variant 2 only. After seed treatment with *Trichoderma* sp., the plant dry weight was significantly increased in variant 1 (*R. solani* inoculum evenly distributed in pot volume) and variant 2 (inoculum only in upper half of pots), but not in variant 3 (inoculum only in lower half of pots).

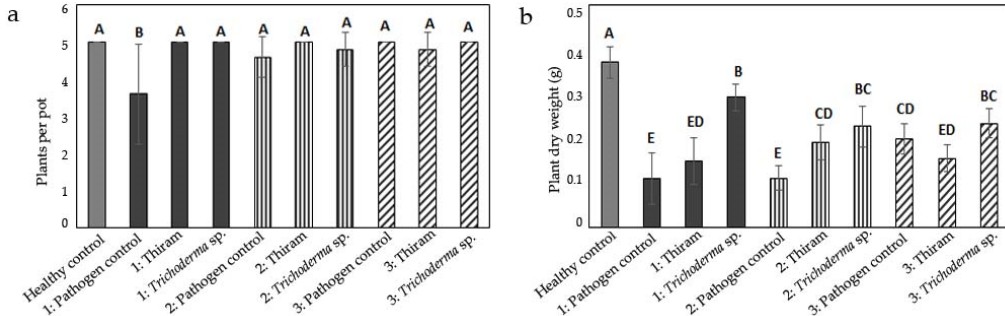

**Figure 3.** Effect of position of inoculum of *Rhizoctonia solani* on (**a**) number and (**b**) biomass of maize seedlings and on efficacy of seed treatment with thiram or a suspension of conidia of *Trichoderma* sp. BI 7376. Variant 1: *R. solani* inoculum mixed homogenously in the entire pot volume; variant 2: *R. solani* inoculum present only in the upper half of pots; variant 3: *R. solani* inoculum present only in the lower half of pots. Means and standard deviation of five pots per treatment with five maize seeds each. Different letters above bars indicate statistically significant differences between the treatments (GLMM; *p* < 0.05).

### 3.4. Influence of Inoculum after Transfer from Healthy Soil

When plants were pre-cultured for 5 days in a layer of non-inoculated substrate and thereafter transferred onto inoculated substrate and further cultivated for 9 days, the plant dry weight was unaffected in the substrate inoculated with *G. ultimum*, but significantly reduced in the pots inoculated with *F. culmorum* and *R. solani* (Figure 4). The habitus of plants was visibly affected (Figure S2).

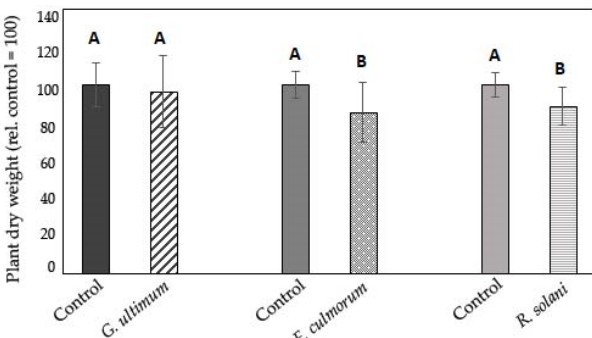

**Figure 4.** Development of maize seedlings transferred from non-inoculated potting substrate onto non-inoculated potting substrate (controls), or from non-inoculated potting substrate onto potting substrate inoculated with *Globisporangium ultimum*, *Fusarium culmorum* or *Rhizoctonia solani*. Means and standard deviation of five pots per treatment with five maize seeds each. Different letters above bars indicate statistically significant differences between the respective control and pathogen (GLMM; $p < 0.05$) based on absolute numbers of plant dry weight per pot.

## 4. Discussion

The aim of the present study was to analyze the impact of the location of the inoculum of three important soilborne fungal pathogens on their pathogenicity towards maize seedlings. The biological control agents *Pseudomonas chlororaphis aurantiaca* BI 7439 and *Trichoderma* sp. BI 7376, as well as the non-systemic fungicide thiram, were included in the experiments. As a seed treatment, dithiocarbamate thiram is mainly used as an anti-*Fusarium* compound. It is, however, currently no longer registered for seed treatment of maize. In the present study, it was included for reasons of comparability with our previous work. *Pseudomonas chlororaphis aurantiaca* BI 7439 and *Trichoderma* sp. BI 7376 have shown activity against soilborne *F. culmorum* comparable to the chemical thiram. In these experiments, *Trichoderma* sp. also protected against *R. solani* [7].

Under the experimental conditions employed, the prevailing disease pattern caused by the pathogens was retarded growth in the case of *R. solani,* and partial pre-emergence damping off with yellowing and impaired growth of the remaining plants in the cases of *G. ultimum* and *F. culmorum*. *G. ultimum* was unique in that disease symptoms were absent in the variant with the inoculum only present in the lower half of the pots. In contrast, *F. culmorum* and *R. solani* appeared to be able to also infect roots at a distance from the seeds. This was also seen in the experiments, where the germination of plants in non-inoculated substrate was followed by transfer to inoculated substrate. Nevertheless, as in the case of *G. ultimum,* damage from *F. culmorum* and *R. solani* was also the largest in the variants where the seeds germinated in the immediate vicinity of the inoculum.

The observations are likely related to the mode of infection of the pathogens. In natural soils, infection of the host can only commence after germination of the resting structures (i.e., oospores and sporidia of *Globisporangium*, chlamydospores of *Fusarium*, and sclerotia of *Rhizoctonia*). Because the inocula used in our study were only 3–6 days old, we assume that resting structures were not involved and the infections were mainly due to outgrowing hyphae that colonized the germinating seeds directly, although detailed studies were not conducted.

Stimulatory effects on fungal spore germination and growth of soluble and volatile substances diffusing from germinating seeds are well documented [19]. Infection of germinating seeds is notably fast by *G. ultimum*. Colonization of seeds of cucumber and the pericarp of sugarbeet seeds by *G. ultimum*, respectively, was observed as early as 4 h after planting [20,21]. Three days after inoculation of germinating maize embryos with *F. moniliforme* (syn. *F. verticillioides*), hyphae of the pathogen were present in the scutellum and the radicle [22]. However, judging by the accumulation of specific proteins, the first host reactions had already occurred 6 h after inoculation [23]. Root infection and colo-

nization by *R. solani* [24,25] and species of *Fusarium* [26] and *Globisporangium* [27,28] has been described for different crops. It is not known why the roots at a distance from the seeds were not affected by *G. ultimum* in our study. A possible explanation could be the growing conditions. Damping off by *G. ultimum* is often associated with high soil moisture conditions and low temperatures [29,30] that are believed to act directly on the host by decreasing host vigor and increasing seed exudation [31]. It is likely that the lack of stress from excessive soil moisture and the ambient temperature of 20 °C favored the development of the maize seedlings relative to the growth of the pathogen. In the experiments with inoculation of the potting substrate with *G. ultimum*, the chemical thiram failed to provide protection, but showed significant activity against soilborne *F. culmorum*. Both observations are in agreement with our previous study [7].

It appears plausible to expect that protective chemical compounds applied to seeds can only exert activity in the spermosphere [32] and in a narrow, extended diffusion zone around the seeds. Therefore, the comparatively good activity of thiram against soilborne *Fusarium* seen here and in our previous experiments may be another indication for the higher pathogenic impact of infection on the germinating seeds compared to the infection of roots at a distance from the seeds. In the experiments with *R. solani*, seed treatment with *Trichoderma* sp. BI 7376 caused significant increases in plant dry weight in the variants with direct contact between the seeds and the inoculum, which, again, points to the relative importance of infection of the germinating seeds. Protection was not significant when the inoculum was located in the lower half of the pots, away from the seeds, despite lower disease pressure in the respective pathogen control. Because biocontrol activity is often associated with the ability to colonize plant surfaces and the rhizosphere [33], this may be an indication of the inability of our *Trichoderma* strain to colonize the roots. However, more results are needed to justify this assumption, and experiments are underway to elucidate the ability of *Trichoderma* sp. BI 7376 to colonize maize roots.

## 5. Conclusions

In tests under controlled conditions with artificial inoculation of the potting substrate, the pathogenic effect of *G. ultimum*, *F. culmorum* and *R. solani* on maize seedlings can be expected to be the strongest if the germinating seeds are in direct contact with the inoculum. For this reason, seed-applied control agents need to be active immediately at the onset of germination in order to be efficacious against these pathogens. For microbial seed treatment agents, this implies that the ability to colonize the spermosphere is of utmost importance. In our experimental system, protection of maize roots against the studied pathogens appeared to be less important than protection of the spermosphere. Clearly, the transferability of our results to open field conditions still needs to be evaluated. There was limited damage by *R. solani* and *F. culmorum*, but the roots were apparently unaffected by *G. ultimum*, despite identical test conditions. Therefore, in order to protect against seedling diseases with microbial agents, the ability to colonize roots at a distance from the germinating seed may not be a universal requirement for microbial agents, but would depend on the pathogen being controlled.

**Supplementary Materials:** The following supporting information can be downloaded at https://www.mdpi.com/article/10.3390/agronomy12061388/s1: Figure S1: Habitus of maize seedlings (**a**) 8 and (**b**) 14 days after sowing in pots with inoculum of *Rhizoctonia solani* located in different positions; Figure S2: Maize seedlings germinated in non-inoculated potting substrate and placed 5 days after planting on non-inoculated potting substrate (control; **left**) or on potting substrate inoculated with *Rhizoctonia solani* (**right**).

**Author Contributions:** Conceptualization: E.K., T.B. (Tim Birr) and A.L.; methodology: E.K., P.Z. and A.L.; investigation: E.K. and P.Z.; validation: T.B. (Tanja Bernhardt); writing—original draft preparation: E.K. and A.L.; writing—review and editing: E.K., P.Z., T.B. (Tim Birr), T.B. (Tanja Bernhardt) and A.L. All authors have read and agreed to the published version of the manuscript.

**Funding:** This research received no external funding.

**Institutional Review Board Statement:** Not applicable.

**Informed Consent Statement:** Not applicable.

**Data Availability Statement:** Data available upon request.

**Acknowledgments:** We thank Astrid von Galen for technical support.

**Conflicts of Interest:** The authors declare no conflict of interest.

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
