# Peer review of "Location of Pathogen Inoculum in the Potting Substrate Influences Damage by Globisporangium ultimum, Fusarium culmorum and Rhizoctonia solani and Effectiveness of Control Agents in Maize Seedlings"

_agronomy, doi:10.3390/agronomy12061388_

Round 1

Reviewer 1 Report

This is a great manuscript, very helpful to anyone working with seedling diseases regardless of crop.  On line 256 please and in to the sentence "present only in the lower half of the pots.  In citation 14 please add S to Screening of antagonistic.  Otherwise no other changes are needed.

Reviewer 2 Report

"The aim of the present study was to analyse the impact of the location of the inoculum of three important soilborne fungal pathogens on their pathogenicity towards maize seedlings": Then why used Biocontrol agents? The condition you get in the field and in pot are different so, how you can measure the effect of inoculum at different depth of the soil? "The results show that the location of the inoculum has an impact on the resulting 79 infection pressure and that the response depends on the pathogen employed": is it not true for all the pathogens? The entry point of all pathogens vary. I got confused about the main objective.

Reviewer 3 Report

Paper reports quite interesting research on the effect of location of pathogen inoculum on damages caused by three pathogens in maize seedlings. Results have great utility value for entities conducting different types of research on the efficacy of plant protection products. Efficient method of inoculation is the crucial factor ensuring reliability of results. It should be emphasized, that the paper is well written, which shows very good organization, readability and grammar. However, some editorial mistakes can be found in the text. So, the manuscript should be improved by the Authors.
